# Evidence of Extensive Circulation of *Yersinia enterocolitica* in Rodents and Shrews in Natural Habitats from Retrospective and Perspective Studies in South Caucasus

**DOI:** 10.3390/pathogens10080939

**Published:** 2021-07-26

**Authors:** Tata Imnadze, Lile Malania, Neli Chakvetadze, Irma Burjanadze, Natalia Abazashvili, Ekaterine Zhgenti, Ketevan Sidamonidze, Ekaterine Khmaladze, Vakhtang Martashvili, Nikoloz Tsertsvadze, Paata Imnadze, Andrei Kandaurov, Ryan J. Arner, Vladimir Motin, Michael Kosoy

**Affiliations:** 1National Center for Disease Control and Public Health, 0186 Tbilisi, Georgia; tataimnadze@gmail.com (T.I.); malanial@yahoo.com (L.M.); science@ncdc.ge (N.C.); irmaepi@gmail.com (I.B.); abazashvilin@yahoo.com (N.A.); eka_zh@hotmail.com (E.Z.); ksidamonidze@gmail.com (K.S.); khmaladze.e@gmail.com (E.K.); vakhtangmartashvili@gmail.com (V.M.); niko@ncfc.ge (N.T.); pimnadze@ncdc.ge (P.I.); 2Faculty of Medicine, Public Health and Epidemiology Department, Ivane Javakhishvili Tbilisi State University, 0179 Tbilisi, Georgia; 3Institute of Zoology, Ilia State University, 0177 Tbilisi, Georgia; a.s.kandaurov@gmail.com; 4Ryan Arner Science Consulting LLC, Freeport, PA 16229, USA; ryan.j.arner@gmail.com; 5Department of Pathology, University Texas Medical Branch, Galveston, TX 77555, USA; vlmotin@utmb.edu; 6KB One Health LLC, Fort Collins, CO 80521, USA

**Keywords:** *Yersinia enterocolitica*, rodents, shrews, bacterial genome, heat-labile toxin, SNP-typing, virulence genes, South Caucasus

## Abstract

*Yersinia enterocolitica* culture-positive rodents and shrews were reported in different territories across Georgia during 14 of 17 years of investigations conducted for the period of 1981–1997. In total, *Y. enterocolitica* was isolated from 2052 rodents (15 species) and 33 shrews. Most isolates were obtained from *Microtus arvalis*, *Rattus norvegicus*, *Mus musculus*, and *Apodemus* spp. During the prospective study (2017−2019), isolates of *Yersinia*-like bacteria were cultured from 53 rodents collected in four parts of Georgia. All the *Yersinia*-like isolates were confirmed as *Y. enterocolitica* based on the API 20E and the BD Phenix50 tests. Whole-genome (WG) sequencing of five rodents and one shrew strain of *Y. enterocolitica* revealed that they possessed a set of virulence genes characteristic of the potentially pathogenic strains of biogroup 1A. All isolates lacked distinguished virulence determinants for YstA, Ail, TccC, VirF, and virulence plasmid pYV but carried the genes for YstB, YmoA, HemPR-HmuVSTU, YaxAB, PhlA, PldA, ArsCBR, and a flagellar apparatus. One strain contained a gene highly homologous to heat-labile enterotoxin, a chain of *E. coli*, a function not previously described for *Y. enterocolitica*. The WG single-nucleotide polymorphism-based typing placed the isolates in four distinct phylogenetic clusters.

## 1. Introduction

*Yersinia enterocolitica*, as a species, belongs to the genus *Yersinia* within the Enterobacteriaceae family that combines *Yersinia pestis*, agent of plague, and a large number of other phylogenetically and phenotypically close species and strains [1,2]. Bacteria assigned to *Y. enterocolitica* represent a heterogeneous and diverse group of strains, but most information about this group arrived from investigations of bacterial cultures obtained from human patients. Because of the origin of these strains, they designated those strains and biogroups as pathogenic. In addition, the pathogenicity of the strains is declared on the presence of some genes (virulence markers), which facilitates colonization of human or animal organisms [3,4]. Since only a limited number of the biogroups obtained from human patients have been investigated, a multitude and diversity of *Y. enterocolitica* strains circulating in nature remain poorly investigated and could be well beyond the scope of the so-called “pathogenic biogroups”. Even among investigated variants, the recognition of pathogenic strains is conditional. For example, strains of biogroup 1A were considered non-pathogenic because of the absence of classical virulence markers until the demonstration of their pathogenic potential [5].

In contrast to *Y. pestis* that causes plague, and similarly to *Y. pseudotuberculosis*, some biovars of *Y. enterocolitica* cause gastrointestinal illness and various clinical manifestations in people, generally termed as ‘yersiniosis’ [3]. The ecology of *Y. enterocolitica* is very different from the ecology of *Y. pestis*. While the plague pathogen, *Y. pestis*, persists in nature through cycles of flea transmission between specific rodents [6,7], *Y. enterocolitica* is widespread in the environment (e.g., soil) and is commonly transmitted via contaminated food [8]. At the same time, there are many reports of detection of this bacterium in various animals that led to the recognition of this species as a zoonotic agent [8]. The most well-known zoonotic association of *Y. enterocolitica* is with domestic pigs since contaminated pork is considered to be the most important source for human yersiniosis. Bacteria of this species have been isolated from other domestic animals (cattle, sheep, and goats) and from numerous species of wild animals (wild boars, deer, bats, foxes, jackals, and raccoons) [9,10,11,12,13].

Much less is known about the distribution of *Y. enterocolitica* in rodents and shrews. Most reports are limited to culturing these bacteria from commensal rats living in close proximity to pig farms [14]. When identified, the isolates obtained from these rats were undistinguished from strains found in pigs. There are reports of isolation of *Y. enterocolitica* strains from small free-living rodents from China, Japan, Czechoslovakia, Finland, Poland, Scandinavian countries, and Crimea [15,16,17,18,19,20]. Nine strains of *Y. enterocolitica* were obtained from common shrews in Finland [18]. The strains of *Y. enterocolitica* identified in natural sources were antigenically variable, which led to referring to some of them as *Y. enterocolitica*-like bacteria [1]. Classification of the bacteria that are close but different from well-accepted *Y. enterocolitica* biovars and await further developments, partially because of the limited representation of *Y. enterocolitica*-like cultures and genomes obtained from animal sources.

Plague has continuously circulated in the Caucasus region for many centuries [21]. Since 1992, scientists in Georgia have continued the investigation of rodents within natural foci of plague. Occasionally, the bacteriologists working in plague foci reported cultures of *Y. enterocolitica* from investigated rodents, but those reports were either unpublished or presented in archived documents which were previously classified material and unreachable to outside specialists. An additional source for information about the presence of *Y. enterocolitica* and related bacteria has emerged after the U.S. Defense Threat Reduction Agency (DTRA) supported a special project to investigate the molecular epidemiology and ecology of *Yersinia* species in the plague endemic territory in Georgia and neighboring Azerbaijan. This paper summarizes results on the prevalence of *Y. enterocolitica* in populations of different rodent species based on (1) retrospective analysis of available data on the identification of *Y. enterocolitica* in rodents during routine plague surveillance conducted across the territory of Georgia and (2) a prospective investigation of rodents captured in a plague focus located in the Eastern part of Georgia.

## 2. Material & Methods

### 2.1. Archived Documents

We have carefully analyzed annual reports of the Republican Anti-plague Station for the period of 1960–1997 archived at the National Center for Disease Control & Public Health (NCDC). Information on the detection of *Y. enterocolitica* in rodents and shrews was found in reports starting in 1981. The reported analysis covers the period from 1981 to 1997. Only common names of rodents have been provided in the reports, and where possible, the names follow current rodent taxonomy with the use of Latin names. Generally, we referred to the local Caucasian guide [22] and the checklist in [23], with some changes according to modern issues [24]. By doing this, we wish to avoid any controversy and debates between zoologists regarding taxonomic revisions. A selection of some Latin names (e.g., *Apodemus* versus *Sylvaemus* for wood mice) was made purely for a conventional purpose and does not reflect our preference for any particular taxonomic scheme. All mice belonging to the genus *Apodemus* were combined together because of the impossibility of species name verification.

### 2.2. Field Investigations

In total, 573 rodents and shrews were collected during 2017–2019 for projects conducted by NCDC. All procedures for capturing and processing small rodents were approved by the Institutional Animal Care and Use Committee at the Georgian Association for Laboratory Animal Science (Protocol No. PA 005/17). A combination of two kinds of the Sherman live traps (Sherman Traps Inc., Tallahassee, FL, USA) were used to capture rodents: 7.6 cm × 8.9 cm × 22.9 cm aluminum folding and non-folding traps collecting for mice and voles and 12.7 cm × 12.7 cm × 38.1 cm Large Non-Folding Heavy-Duty traps to target rats and jirds. The traps were set out with intervals of approximately 4–6 m during daylight hours, usually late afternoon, over a 1–2-day period for each trap line. The small mammals were collected from four sites located in different regions of Georgia: (1) Dedoplistskaro in the Kakheti region of eastern Georgia close to Azerbaijan; (2) Aspindza in Samtskhe-Javakheti region of southern Georgia; (3) Marneuli in the Kvemo Kartli region in south-eastern Georgia; and (4) near Batumi City located on the Black Sea coast in Ajaria of southwestern Georgia.

Prior to blood and ectoparasite collection, the captured small mammals were anesthetized using open-drop exposure to Isoflurane (Aesica, Queenborough, UK) using a 20% volume/volume mixture for mice and voles and a 30% volume/volume mixture for rats and jirds. Blood samples from mice, voles, and small jirds were collected by bleeding from the retro-orbital plexus and by femoral venipunctures from medium-size animals (large jirds and rats). Rodents were then euthanized via either exsanguination or exposure to a lethal dose of Isoflurane. Tissue samples (blood, spleen, liver, and intestine) were sterilely collected from each rodent and placed in separate pre-labeled cryogenic vials.

### 2.3. Microbiology

The laboratory procedures for culturing *Yersinia* bacteria were conducted at the Richard Lugar Center for Public Health Research, Tbilisi, Georgia, following the standard operating procedures approved by NCDC’s QMS committee (SOP-GBG -116- Identification of *Yersinia* species _V3.0, 2020). Suspensions of small intestine samples were inoculated into 1% peptone water followed by growth for 28 days at 4 °C. The inoculum was plated every 5th day on Endo and CIN agar (bioMérieux, Marcy-l’Étoile, France), and plates were incubated at 28 °C for 7 days. Bacterial colonies with a morphology resembling *Yersinia* (round shape, 0.5–1 mm diameter size, and grayish-pinkish color; a “bulls-eye” appearance on CIN agar) were sub-cultured on Endo agar plates at 28 °C for 24–48 h. The colonies obtained were analyzed microscopically after Gram staining. BD Phoenix (Becton & Dickinson, Eysins, Switzerland) and API (Analytical Profile Index, bioMérieux, France) tests were used according to the manufacturers’ guidance for further analysis. Identification of *Y. enterocolitica* was conducted with the use of the API20E test system (Analytical Profile Index, bioMérieux, France) and the BD Phenix50 Automatic System using a Gram-Negative ID Panel (Becton & Dickinson, Switzerland) according to the manufacturers’ guidance for further analysis.

### 2.4. Determination of Plasmid Content

The plasmid profile of *Y. enterocolitica* isolates was determined by the phenol method [25] as described by us previously [26]. The vaccine strain of *Y. pestis*, EV76, containing three plasmids of known molecular weight (101 kb, 70.5 kb, and 9.5 kb), was used as the plasmid size standard.

### 2.5. Selection of Y. enterocolitica Isolates for Sequencing

The isolates chosen for the WG sequencing are listed in Table 1. They were selected from different geographical locations across the country (Figure 1) and were isolated from different rodent species and a shrew during the same year. Three of these strains were isolated from rodents collected in Dedoplistskaro, Kakheti Region, during the investigation of the active plague epizootic.

### 2.6. Genome Sequencing and Assembly

DNA for WG sequencing of six selected *Y. enterocolitica* strains were prepared using the QIAGEN Genomic DNA Kit (QIAGEN, Valencia, CA, USA). Extracted DNAs were quantified using a Qubit DNA HS Quantitation kit (Invitrogen, Thermo Fisher Scientific, Waltham, MA, USA) on a Qubit 3.0 fluorometer. Sequencing libraries were prepared using the NEBNext^®^ Ultra DNA library prep kit for Illumina (New England Biolabs, San Diego, CA, USA) according to the manufacturer’s instructions. DNA (1 g) was sheared by sonication using a Covaris M220 sonication system (Covaris, Inc., Woburn, MA, USA) to an average of 500 bp fragment size. For library quantification, a Qubit dsDNA HS kit (Invitrogen) was used. Fragment size distribution was verified with a high sensitivity DNA chip on a Bioanalyzer (Agilent Technologies, Inc. Germany). Libraries were pooled and sequenced on an Illumina MiSeq instrument in paired-end mode 2 × 251 nt (500 cycles) using a Miseq reagent kit V2.

This Whole Genome Shotgun project has been deposited at DDBJ/ENA/GenBank (Bioproject PRJNA719294) under the accession JAGJTI000000000 (18E17 065 D I-33), JAGJTJ000000000 (LcENCDC Rint 5), JAGJTK000000000 (18ENCDC AsAtc Rint-11-12), JAGKRV000000000 (18E17 065 D Rint-95), JAGKRW000000000 (18E17 065 D Rint-91), and JAGKRX000000000 (B18 Rint-231). The versions described in this paper are versions JAGJTI010000000, JAGJTJ010000000, JAGJTK010000000, JAGKRV010000000, JAGKRW010000000, and JAGKRX010000000.

### 2.7. Genome Sequence Processing and Phylogenetic Analyses

Obtained raw sequence reads were processed using the CLC Genomics Workbench 12.0.3 (CLC Bio, Aarhus, Denmark). After quality trimming, the sequence reads were de novo assembled (k-mer size = 20). Contigs generated in CLC were uploaded into the EDGE bioinformatics platform version 2.3.1 and processed using the following analytical workflows: reference-based analysis, annotation (Prokka), taxonomy classification, and gene family analysis. For comparison against known references, RefSeq complete genomes of *Y. entercolitica* were selected from the EDGE drop-down list. The virulence genes were identified using ShortBRED with a database generated by the developers of EDGE using VFDB.

Illumina sequence reads were also uploaded into the integrated software environment EnteroBase (http://enterobase.warnick.ac.uk) under the *Yersinia* database. Sequence reads were processed with the EnteroBase pipeline, including the sequence read quality check, assembly (SPAdes), and annotation. The six uploaded *Y. enterocolitica* isolates were compared to a publicly available whole-genome sequence of selected 125 *Y. enterocolitica* strains in the database using an in silico SNP-typing scheme module. Maximum likelihood phylogenetic trees of core, non-repetitive SNPs were called against a reference draft genome (strain 1127) and constructed using the inbuilt dendrogram option of SNP Projects.

## 3. Results

### 3.1. Prevalence of Y. entercolitica in Rodents and Shrews Based on Archived Records

During 1981−1997, *Y. enterocolitica* culture-positive rodents and shrews were reported during 14 of 17 years of the investigations. In total, *Y. enterocolitica* bacteria were isolated from 2052 rodents of 14 species (Table 2). Of those, most isolates were obtained from common voles *Microtus arvalis* (*n* = 1386), Norway rats *Rattus norvegicus* (*n* = 198), wood mice *Apodemus* spp. (*n* = 180), and house mice *Mus musculus* (*n* = 131) (Table 2).

Infected rodents and shrews were identified in different territories across Georgia, but rodent species carrying *Y. enterocolitica* varied. Infected commensal rodents, such as *R. rattus*, *R. norvegicus*, and *M. musculus*, were found in Tbilisi (capital of Georgia) and in seaports on the Black Sea coast (Batumi, Poti, and Sukhumi). Most infected common voles were found in southern and southwestern regions of Georgia, while infected social voles (*Microtus socialis*) and Libyan jirds (*Meriones libycus*) were captured in the steppe foothills of Eastern Georgia (Figure 2). Interestingly, there was a report of isolation of *Y. enterocolitica* from an Oriental rat flea *Xenopsilla cheopis* collected from *R. norvegicus* from the city of Tbilisi in 1987.

### 3.2. Isolation of Y. enterocolitica and Y. enterocolitica-Like Organisms during the Perspective Study

Growth of bacterial colonies morphologically suspected as *Yersinia* was observed after five days of incubation of 53 out of 572 freshly collected intestines from rodents and shrews randomly selected for this study. The characteristics of the colonies on CIN agar included a deep red center with transparent margins (a “bulls-eye” appearance). Gram-stained microscopy showed gram-negative bacillar-shaped microorganisms. All *Yersinia*-like isolates obtained from rodents were identified as *Y. enterocolitica* or *Y. enterocolitica*-like bacteria based on the API20E tests. Positive samples were obtained from *R. norvegicus* (7/63), *M. musculus* (10/36), Macedonian mice, *M. macedonicus* (4/85), *Microtus socialis* (5/39), *M. arvalis* (2/10), *Apodemus* spp. (21/187), bicolored shrews *Crocidura leucodon* (4/41), and from an unidentified, white-toothed shrew of the genus *Crocidura* (presumably, a lesser shrew *C. suaveolens* based on the habitat characteristics, but a complete identification cannot be confirmed).

### 3.3. Phenotypical Characterization of Freshly Isolated Y. enterocolitica Strains

Gram-stained microscopy showed gram-negative bacillar-shaped microorganisms. A bacterial suspension was used to rehydrate each of the wells, and the strips were incubated. All positive and negative API test results were compiled to obtain a profile number, which was then compared with profile numbers in a commercial codebook. All 53 *Yersinia*-like isolates were identified as *Y. enterocolitica* based on the API 20E tests and the BD Phenix50 Automatic System. The phenotypic API20E characteristics of 53 *Y. enterocolitica* isolates are shown in Appendix A.

Out of six isolates selected for sequencing, three had identical biochemical properties, strains 18ENCDC AsAtc Rint 11-12, 18E17 065 D I Rint-91, and 18E17 065 D I Rint-95. The latter two were isolated from the plague focus in Dedoplistskaro from different mouse species (Table 1). Another strain from this region isolated from a vole, 18E17 065 D I-33, differed in one position. Two other isolates, LcENCDC Rint 5 and B18 Rint -231 differed in two non-identical positions.

### 3.4. Genome Size and Sequences of Y. entercolitica Strains Isolated from Small Mammals

The sizes of the assembled genomes of *Y. enterocolitica* strains 18ENCDC AsAtc Rint-11-12, LcENCDC Rint 5, 18E17 065 D I-33, 18E17 065 D I Rynt-95, 18E17 065 D I Rynt-91, B18 Rint-231 were 4,673,420 bp (40 contigs), 4,696,602 bp (45 contigs), 4,899,686 bp (91 contigs), 4,872,315 bp (75 contigs), 4,871,034 bp (95 contigs), and 4,775,715 bp (124 contigs), respectively. The genome properties are shown in Table 3.

### 3.5. Virulence Genes of Sequenced Y. entercolitica Isolates

The analysis of virulence genes showed that all isolates did not possess genes encoding heat-stable enterotoxin A YstA, adhesin Ail, insecticidal toxin TccC, virulence plasmid pYV containing T3SS, and virulence regulator VirF (Appendix A). In contrast, all six strains had heat-stable enterotoxin B YstB, a modulator of expression for virulence functions YmoA, ferric enterobactin transport system FepBDGC, direct heme uptake system HemPR-HmuVSTU, pore-forming cytotoxin YaxAB, phospholipases PhlA and PldA, arcenic cluster ArsCBR, flagellar apparatus, and type 2 secretion system.

Both present and absent virulence functions pointed out that these isolates belong to the potential pathogenicity *Y. enterocolitica* biogroup A1. Nevertheless, the isolates differed from each other in a number of virulence proteins. Notably, only three strains contained classical invasin Inv, typically located within the flagellar cluster in *Y. enterocolitica*. Nevertheless, all strains, including those missing the Inv function, had putative invasin-like proteins with low homology to Inv. Three strains isolated in the same region (Dedoplistskaro) did not contain MyfA fimbrial protein, which is an adhesin essential for virulence and in vivo-expressed subtilisin/kexin-like protease HreP, RtxA-like putative leukotoxin, and cytolethal distending toxin subunit B-like protein CdtB. The strains isolated in other regions all contained these four virulence factors. Moreover, the Dedoplistskaro isolates contained tRNA(fMet)-specific endonuclease VapC, which was absent in the genomes of all other strains.

Two strains from Dedoplistskaro, 18E17 065 D I Rint-91 and 18E17 065 D I Rint-95, isolated in close proximity, although from different animal species (Table 1, Figure 1), were identical with regards to the possession of virulence genes. In contrast, the third strain from this region (18E17 065 D I-33) isolated distantly from the former two strains lacked ADP-rybosyltransferase Pertussis-like toxin YtxA, the function identified in all our strains. Finally, this strain contained a distinct virulence protein, YltA, heat-labile enterotoxin, a chain with significant homology to LTA toxin of enterotoxigenic *E. coli* and other microorganisms, such as *Providencia alcalifaciens*, *Cronobacter*, and *Citrobacter sp.* S-77 (Appendix A). The unique feature of 18ENCDC AsAtc Rint-11-12 isolate was an absence of VapBC type II toxin-antitoxin system, and the B18 Rint-231 isolate lacked critical components of chromosomal T3SS; both functions were found in all other strains.

### 3.6. Plasmids

The screening of *Y. enterocolitica* isolates, from small mammals, by gel electrophoresis revealed the presence of plasmids in all three isolates from Dedoplistskaro (Appendix A), while other isolates had no plasmids. Both strains 18E17 065 D I Rint-91 and 18E17 065 D I Rint-95 that had identical sets of virulence genes and phenotypic properties by API20E showed the same plasmid profile. We were not able to identify plasmid contigs in the sequenced genomes since these plasmids were large and probably had low copy numbers. The crude estimate of their molecular weight, based on the standards from *Y. pestis* vaccine strain EV76, suggested that the sizes of the large plasmids were about 200 kb, 90 kb, and 60 kb.

Another Dedoplistskaro isolate, 18E17 065 D I -33, likely had similar 200 kb and 60 kb plasmids and lacked the 90 kb plasmid. Nevertheless, we were able to identify contigs associated with the small multi-copy number plasmids in 18E17 065 D I Rint-91 and 18E17 065 D I Rint-95 isolates. The plasmids from both strains were identical except for a small deletion of 29 bp found in 18E17 065 D I Rint-91. The size of the plasmid was 4.5 kb, and it contained replication initiation protein repE_1, two short hypothetical proteins, and to our surprise, the type 4 secretion system (T4SS) VirB6 conjugal transfer protein. This protein has significant homology with other VirB6 proteins found in *Yersinia pseudotuberculosis*, *Yersinia kristensenii*, *Yersinia massiliensis*, *Klebsiella pneumoniae* NCTC9632, *E. coli*, and *Salmonella enterica* PNUSAS047895 (Appendix A).

### 3.7. Phylogenetic Relationship

To define genetic relationships among the small mammal *Y. enterocolitica* isolates from Georgia and worldwide isolates of *Y. enterocolitica* strains, 125 core genome SNP-typing was performed using the EnteroBase platform. We included the isolates of *Y. enterocolitica* obtained from bats in Georgia [26] in this analysis. The results of the SNP-typing are shown in Appendix A. As expected from the genomic analysis of the presence of virulence genes, all the isolates obtained from Georgian small mammals belonged to the biogroup 1A.

The three strains isolated from Dedoplistskaro grouped together and formed a separate cluster, which was quite distant from the rest of *Y. enterocolitica* isolates in the entire database. The strains 18E17 065 D I Rint-91 and 18E17 065 D I Rint-95 had identical SNP-types, despite being isolated from rodents captured in close proximity, though from different rodent species (*M. macedonicus* and *Apodemus* sp.). This suggests a potential transmission of this variant between mice of different species. Interestingly, the infected mice were identified within a plague-endemic territory where *Y. pestis* was recently detected in wild rodents and their fleas (unpublished data). The third strain (18E17 065 D I-33) from the same territory (Dedoplistskaro) was isolated from a vole *M. socialis* and was phylogenetically distinct from the above-mentioned two strains. There was only one strain in the entire EnteroBase, which belonged to the same cluster as the Dedoplistskaro strains, the strain Y18EE025 of an undetermined biogroup and serotype. Interestingly, the SNP-type of the strain 18ENCDC AsAtc Rint-11-12 isolated from *R. norvegicus* was identical to the *Y. enterocolitica* strains obtained from dead bats described in Georgia [26].

This finding suggests a wide distribution of this genotype of *Y. enterocolitica* in this part of Georgia. The SNP-type of the strain (B18 Rint -231) isolated from a shrew (Adilia, suburb of Batumi, Black Sea port) was also unique in the entire database. The only strain grouped together on tree 18005024 of biogroup 1Awas an undetermined human isolate serotype from Luxemburg, 2018. Finally, the LcENCDC Rint 5 strain isolated from *R. norvegicus* in another region in Southern Georgia also had a distinct SNP profile and grouped together with the strain YE13/03 of biogroup 1A, serotype O:6,30 isolated in the United Kingdom from human feces in 2003 (BioSample: SAMEA980154; SRA: ERS008611).

## 4. Discussions

Until recently, frequent reports of isolation of strains, which were identified as *Y. enterocolitica* from wild rodents collected in plague endemic areas in the Caucasus and Central Asia, remained undocumented in English language literature. The Soviet Union’s Anti-plague system covered a vast territory of these regions. Zoologists and bacteriologists working for the Anti-plague stations captured and processed an enormous number of rodents in search of plague and other rodent-borne diseases. According to the laboratory procedure for the identification of plague pathogens, bacteriophage typing was routinely applied for bacteria morphologically identified as *Yersinia*. Bacteria of *Y. pestis* are sensitive to specific bacteriophage lysis at 20 °C, whereas bacteria of *Y. enterocolitica* do not lyse at temperatures below 28 °C [27].

Isolation of numerous cultures of *Y. enterocolitica* from various species of wild mammals in Georgia strongly suggests a wide circulation of this species of *Yersinia* in natural habitats across all landscape zones of the Caucasian region. This discovery was unexpected because previously, *Y. enterocolitica* has been primarily known as a human enteric pathogen with some involvement of domestic animals (pigs) with rare reports from commensal rats living closely to pig farms. Therefore, it was not entirely surprising that several specimens of *R. rattus*, and house mice *M. musculus,* were found culture-positive for *Y. enterocolitica*. Unexpected was a high occurrence of this bacterium in wild rodents inhabiting various natural habitats across the region.

During the perspective study organized specifically to identify *Yersinia* species in 14 rodents of different species, two species of voles (*M. socialis* and *M. majori*) and two species of wild mice (*A. uralensis* and *A. witherbyi*) were found to be infected with this bacterium. These rodents commonly occupy natural habitats far from human settlements and pig farms. Noticeably, four Caucasus field mice (*A. ponticus*), which were caught in a forest near a small pig farm where pigs reportedly grazed under the forest cover, were tested negative for *Y. enterocolitica*.

Another important question has arisen about the interaction between bacteria of two species belonging to the genus *Yersinia* (*Y. pestis* and *Y. enterocolitica*), which circulate in the same populations of rodents, which requires future investigations. Interestingly, Fukushima et al. (2001) proposed a role of *Y. enterocolitica* as a possible barrier against *Y. pestis* based on the investigation of rodents in a natural plague focus in Ningxia, China. The co-circulation of bacteria belonging to the genus of *Yersinia* in the same rodent populations within plague endemic territories of Georgia presents an excellent opportunity to address this question.

The results from this investigation and few recent publications [28,29] have challenged the dominant perception of *Y. enterocolitica* as a pathogen of people and domestic animals. Cultures obtained provide material for further genetic and bacteriological characterization of isolates of *Y. enterocolitica* and to compare them with human isolates. Such a comparative analysis can help to evaluate the risk for public health of the strains of *Y. enterocolitica* carried by wild rodents.

The phylogenetic analysis revealed that the small mammal *Y. enterocolitica* strains from Georgia belonged to the potentially pathogenic biogroup 1A [3,5,30]. The six isolates investigated here were quite heterogeneous with regard to the presence of different virulence genes, except two mouse strains 18E17 065 D I Rint-95 and 18E17 065 D I Rint-91 from the Dedoplistskaro region, which also contained identical plasmid and biochemical profiles (Appendix A). These two strains were isolated in close proximity from two different mouse species, *A. witherbyi*, and *M. macedonicus*, respectively, and were generally indistinguishable by WG sequencing. The vole (*M. socialis*) strain from this area 18E17 065 D I-33 had an almost identical set of virulence determinants, only missing YtxA Pertussis-like toxin and two of four plasmids characteristic to the former strains.

The interesting feature of all three Dedoplistskaro isolates was a lack of fimbrial adhesin MyfA, suggesting a reduced intestinal colonization potential [5]. Nevertheless, the vole strain 18E17 065 D I-33 contained a gene encoding the A chain of heat-labile enterotoxin (LTA), a function that was not previously seen in *Y. enterocolitica* and rarely found in pathogens other than enterotoxigenic *E. coli* (Appendix A). Since LTA was shown to enhance enteric pathogen adherence and intestinal colonization [31], this may compensate for the lack of the MyfA adhesin in 18E17 065 D I-33 isolate. However, a hallmark of the heat-labile enterotoxin in *Escherichia coli* is its remarkable ability to contribute to diarrhea in travelers, children, and young animals [32]. It remains to be seen whether this *Y. enterocolitica* isolate can produce a holotoxin, where the LTA is assembled with the receptor binding B-subunits (LTB) to promote diarrhea in animal models. The 18E17 065 D I-33 genome search did not reveal obvious candidates for the gene encoding the LTB, which may have a low homology with the LTB from *E. coli* and a distinct receptor binding specificity.

Overall, all six small mammal bacteria strains (*Y. enterocolitica*) from Georgia, as other typical biogroup A1 strains, were deficient in essential virulence factors, such as heat-stable enterotoxin A YstA, adhesin Ail, insecticidal toxin TccC, T3SS from the virulence plasmid pYV, and virulence regulator VirF (Appendix A). Nevertheless, all strains carried the genes for heat-stable enterotoxin B YstB, a modulator of expression for virulence functions YmoA, ferric enterobactin transport system FepBDGC, direct heme uptake system HemPR-HmuVSTU, pore-forming cytotoxin YaxAB, phospholipases PhlA and PldA, arcenic cluster ArsCBR, flagellar apparatus, and a type two secretion system. The isolates varied in the presence of invasin Inv, adhesin MyfA, subtilisin/kexin-like protease HreP, RtxA-like putative leukotoxin, endonuclease VapC, Pertussis-like toxin YtxA and cytolethal distending toxin subunit B-like protein CdtB. The differences in the virulence gene composition suggested a marked diversity of small mammal *Y. enterocolitica* infections in Georgia, although clonal domination can be possible as indicated by the similarity of the strains from Dedoplistskaro, which were isolated from different rodent species.

The diversity of small mammal *Y. enterocolitica* infections in Georgia is clearly seen on the whole-genome SNP-based phylogenetic tree (Appendix A). The strains belonged to four distant clusters with little or no other *Y. enterocolitica* strains from the Enterobase present in such clusters. The exception was the strain LcENCDC Rint 5 isolated from *R. norvegicus* in South Georgia, which grouped with several strains of biogroup 1A. This Georgian isolate was phylogenetically close to the strain YE13/03, serotype O:6,30 isolated in the United Kingdom from human feces in 2003. Interestingly, the strain 18ENCDC AsAtc Rint–11-12 isolated from *Rattus norvegicus* formed a distinct and unique cluster with *Y. enterocolitica* isolates from dead bats in Georgia [26]. Despite the distance between the two sites of isolation being only about 100 kilometers (Figure 1), the two places are separated by mountains, likely preventing a direct communication of the rat and bat populations between the areas. Nevertheless, this might be another example of a wide clonal distribution of *Y. enterocolitica* variants in certain parts of Georgia.

Finally, the most intriguing finding was the observation that *Y. enterocolitica* isolated from different animal species in the natural plague focus of the Dedoplistskaro region were grouped together and might form a different clade or even biogroup in the future. The cluster containing these isolates did not possess *Y. enterocolitica* of a known biogroup, although the set of virulence genes suggests that these isolates are closely related to biogroup 1A. The role of ongoing outbreaks of *Y. pestis* in this area for the selection of these unique *Y. enterocolitica* variants will need further investigation. Overall, the reported observations contribute to a better understanding of the potential public health importance of *Yersinia* species of small mammals beyond plague [10,18,26,32].

## Figures and Tables

**Figure 1 pathogens-10-00939-f001:**
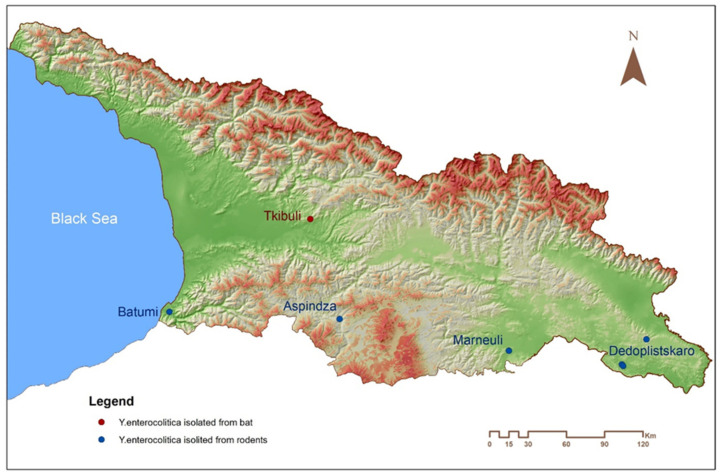
Location of the places of isolation for *Y. enterocolitica* strains from small mammals sequenced in this study: 18E17 065 D I Rint-95 (Dedoplistskaro), 18E17 065 D I Rint-91 (Dedoplistskaro), B18 Rint-231 (Batumi), LcENCDC Rint 5 (Marneuli), 18ENCDC AsAtc Rint-11-12 (Aspindza), and 18E17 065 D I-33 (Dedoplistskaro).

**Figure 2 pathogens-10-00939-f002:**
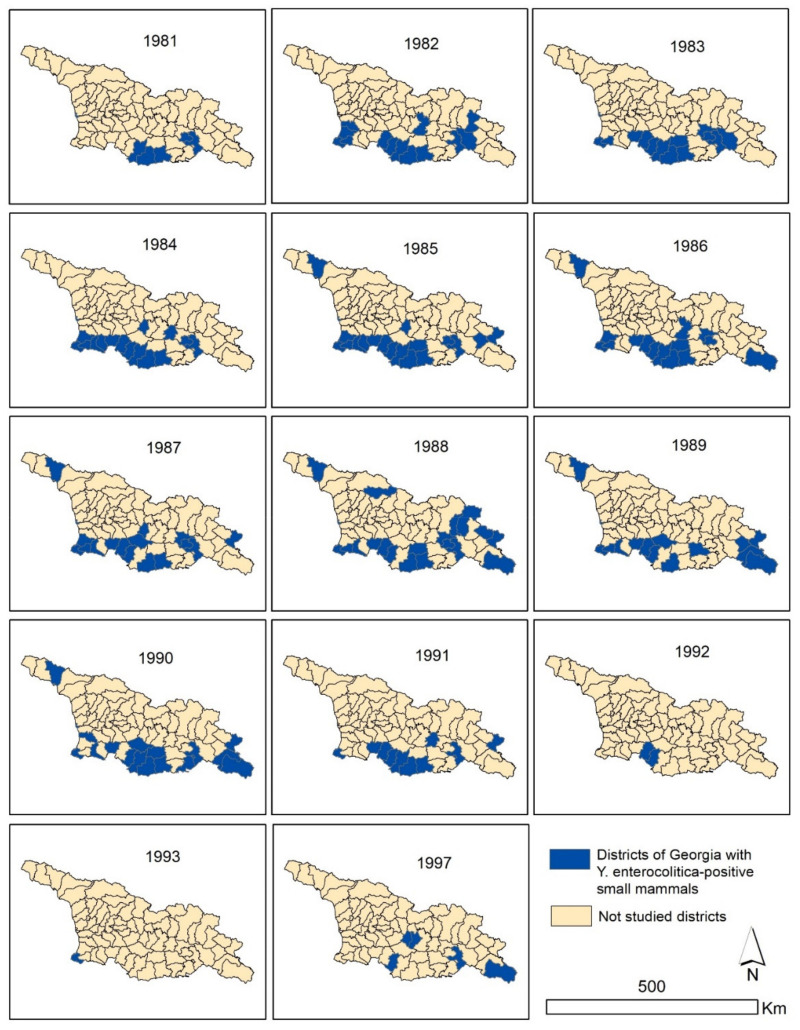
Detection of *Y. enterocolitica* culture-positive rodents by districts of Georgia from 1981 to 1997.

**Table 1 pathogens-10-00939-t001:** List of *Y. enterocolitica* strains sequenced in this study.

ID.	Tissue	Host Species	Location	Year
18ENCDC AsAtc Rint-11-12	Intestine	*Rattus norvegicus*	Aspindza	2018
LcENCDC Rint 5	Intestine	*Rattus norvegicus*	Marneuli	2018
18E17 065 D I-33	Intestine	*Microtus socialis*	Dedoplistskaro	2018
18E17 065 D I Rint-95	Intestine	*Apodemus witherbyi*	Dedoplistskaro	2018
18E17 065 D I Rint-91	Intestine	*Mus macedonicus*	Dedoplistskaro	2018
B18 Rint-231	Intestine	Shrew (*Crocidura* sp.)	Batumi	2018

**Table 2 pathogens-10-00939-t002:** Number of isolates of *Yersinia enterocolitica* bacteria obtained from rodents and shrews collected in Georgia from 1981–1997.

	1981	1982	1983	1984	1985	1986	1987	1988	1989	1990	1991	1992	1993	1997	Total
*Apodemus* spp. *Arvicola terrestris*		1	8	25 4	21 1	32 3	69	13	2	3 3	5			1	180 11
*Chionomys nivalis*					4										4
*Chionomys roberti*				7											7
*Cricetulus migratorius*			1		1				1						3
*Dryomys nitedula*						1									1
*Meriones libycus*						1	2	8	1		2			1	15
*Mesocricetus brandti*		5	1	2			2				2				12
*Microtus arvalis*	8	76	203	165	266	130	116	88	123	73	121	17			1386
*Microtus majori*		10	1	2		3	1	5	1						23
*Microtus socialis*					14	3	2	39	7	5				3	73
*Mus musculus*	1	1	11	21	11	16	11	18	8	15	7		1	10	131
*Rattus rattus*		4				2	1			1					8
*Rattus norvegicus*	3	6	19	20	19	51	38	10	4	17	11				198
Soricidae spp.	1	4	1	6	5	1	5	5		3	1				32
**Total**	**13**	**107**	**245**	**252**	**342**	**243**	**247**	**186**	**147**	**120**	**149**	**17**	**1**	**15**	**2084**

Common names of small mammals: *Apodemus* spp. include Ural field mouse (*A. uralensis*), Steppe field mouse (*A. witherbyi*), and Caucasus field mice (*A. ponticus*); *Arvicola terrestris*—water vole; *Chionomys nivalis*—snow vole; *Chionomys roberti*—Robert’s snow vole; *Cricetulus migratorius*—Grey dwarf hamster; *Dryomys nitedula*—Forest dormouse; *Meriones libycus*—Libyan jird; *Mesocricetus brandti*—Brandt’s hamster; *Microtus arvalis*—Common vole; *Microtus majori*—Major’s pine vole; *Microtus socialis*—Social vole; *Mus musculus*—House mouse; *Rattus rattus*—Black rat; *Rattus norvegicus*—Norway rat; Soricidae—a family of shrews (unidentified species).

**Table 3 pathogens-10-00939-t003:** Genome properties of *Yersinia enterocolitica* strains isolated from small mammals in Georgia.

Genomic Properties	18ENCDC AsAtc Rint-11-12	LcENCDC Rint 5	18E17 065 D I-33	18E17 065 D I Rint-95	18E17 065 D I Rint-91	B18 Rint-231
Assembly size (bp)	4,673,420	4,696,602	4,899,686	4,872,315	4,871,034	4,775,715
N contigs	40	45	83	75	95	124
G+C content (%)	47.06	47.29	46.41	46.5	46.46	46.96
Number of protein-coding genes (CDS)	4218	4258	4549	4455	4453	4392
CDS (Function assigned)	2894	2916	2920	2940	2938	2789
CDS (Hypothetical/putative)	1324	1342	1629	1515	1515	1603
Genes	3103	3132	3124	3144	3142	3001
Number of tRNAs	66	63	69	69	69	72
rRNAs (23S, 16S, and 5S)	6	4	9	5	5	7

## Data Availability

Not applicable.

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
