# Peer review of "Evidence of Extensive Circulation of Yersinia enterocolitica in Rodents and Shrews in Natural Habitats from Retrospective and Perspective Studies in South Caucasus"

_pathogens, 2021, doi:10.3390/pathogens10080939_

Round 1

Reviewer 1 Report

The problem dicussed in the article is interesting and can enrich the knowledge about Y. enterocolitica, especially those belonging to biotype 1A.

Authors often report in their paper the presence of Y. pestis in the study area. It is pity they were not additionally investigate  presence of Y.  pseudotuberculosis , which is more closely related to Y. pestis than to Y. enterocolitica  and is believed to be associated with rodents.

The Y. enterocolitica isolates described in the paper  belonged to biogroup 1A and the authors described them as "low-pathogenic". I cannot agree with that.  Among the biotypes of Y. enterocolitica , biotypes 2, 3, 4, and 5 are referred to as low-pathogenic (some also mention biotype 6), in contrast to biotype 1B, the strains of which are referred as to high-pathogenic. Strains belonging to biotype 1A are considered as non-pathogenic, possibly conditionally-  or potentially-pathogenic if they posses virulence markers. Please use a term other than low-pathogenic as compared to isolates of 1A biotype.

In many parts of the manuscript, the cited literature is not uniformly written.

Line 159 - there is: Imnaze et. al 2020, instead [26]

Lines 381-382 - should be [3, 5, 30], instead the names of authors

Line 393 - should be [31]

Line 399 - also [31], instead Dune et. al.

Listed 28 and 29 literature positions mentioned in the references list are nowhere cited in the manuscript.

Lines 210-211 - probably there is lack of  "were"  in the sentence between "Y. enterocolitica" and  "varied"

Lines 259-260 - If the Authors write about a gene, its name should be in italics and in lowercase, e.g. ystA gene encodes the enterotoxin YstA.

Author Response

Responses to the reviewer 1

The problem discussed in the article is interesting and can enrich the knowledge about Y. enterocolitica, especially those belonging to biotype 1A.

  • Thank you

Authors often report in their paper the presence of Y. pestis in the study area. It is pity they were not additionally investigate presence of Y.  pseudotuberculosis, which is more closely related to Y. pestis than to Y. enterocolitica  and is believed to be associated with rodents.

  • pseudotuberculosis bacteria were sporadically cultured from rodents during previous investigations conducted in Georgia, but during the perspective study the number of such isolates was very limited for an analysis. We hope that we will be able to report more results about detection, identification, and characterization of Y. pseudotuberculosis later.

The Y. enterocolitica isolates described in the paper  belonged to biogroup 1A and the authors described them as "low-pathogenic". I cannot agree with that.  Among the biotypes of Y. enterocolitica , biotypes 2, 3, 4, and 5 are referred to as low-pathogenic (some also mention biotype 6), in contrast to biotype 1B, the strains of which are referred as to high-pathogenic. Strains belonging to biotype 1A are considered as non-pathogenic, possibly conditionally-  or potentially-pathogenic if they posses virulence markers. Please use a term other than low-pathogenic as compared to isolates of 1A biotype.

  • The term was changed for “potentially-pathogenic” instead of “low-pathogenic”.

In many parts of the manuscript, the cited literature is not uniformly written.

  • The references were corrected throughout the text.

Line 159 - there is: Imnaze et. al 2020, instead [26]

  • Corrected

Lines 381-382 - should be [3, 5, 30], instead the names of authors

  • Corrected

Line 393 - should be [31]

  • Reference [5]

Line 399 - also [31], instead Dune et. al.

  • Corrected

Listed 28 and 29 literature positions mentioned in the references list are nowhere cited in the manuscript.

  • The references were added

Lines 210-211 - probably there is lack of  "were"  in the sentence between "Y. enterocolitica" and  "varied"

  • “Species varied” (no change is made)

Lines 259-260 - If the Authors write about a gene, its name should be in italics and in lowercase, e.g. ystA gene encodes the enterotoxin YstA.

  • The sentence was changed with using italics for genes, but without italics for proteins.

Reviewer 2 Report

The manuscript is well written, the introduction sufficiently presents the topic, the results are showed clearly and comprehensibly. The manuscript describes Yersinia enterocolitica strains isolated from rodents and shrews in Georgia and the obtained results further the knowledge concerning circulation and possible reservoirs of this bacterium. In my opinion, this research paper is worth publishing in the target journal. I recommend some minor changes - please see the attachment.

Author Response

Responses to the reviewer 2

Line 9-15: I think that there should be space between the number and affiliation (like in line 8).

      -     Corrected

Line 8-10: is the part „(Country)” necessary? Also in lines 8 and 10, there is an unnecessary

dot at the end

  • This practice is commonly used for avoiding a confusion with the US state of Georgia.

Line 34: if I am not mistaken this is the first occurrence of full name of the bacteria in the text

and it should be followed by “(Y. enterocolitica)

  • When the name of Yersinia enterocolitica is used in the text first time, the full name is provided following Y. enterocolitica after that.

Line 64-66: I advise rephrasing the sentence. For example - “Reports of isolation of Y.

enterocolitica strains from free-living small rodents were from originates form, among others,

China, Japan, Czechoslovakia, Finland, Scandinavian countries, and Crimea”. There are more

reports of isolation from other countries (for example number 32 in the reference list)

  • The sentence was rephrased and the reference was added.

Line 91-92: I recommend rephrasing “Information on the detection of Y. enterocolitica in

rodents and shrews was found in reports starting from 1981 and the analysis covers the period

from 1981 to 1997”

  • The sentence was changed

Line 107-109: According to instructions for authors all units should be converted to SI units.

Inch is not a SI unit

  • Inches were converted to centimeters

Line 107, 118: please check if the country of the producer is necessary (Sherman Traps and

Aesica Queenborough)

  • The country information is added

Line 131: please add producer of Endo and CIN agar

  • The agar producer was provided

Line 130, 131, 134, 349, 350: please unify the Celsius degrees – in lines 131, 349 and 350

there is no space and the symbol seems to be lower as compared to lines 130 and 134

  • Corrected

Line 162-172: I think that there may be some mistake in product/producer information. Some

parts are highlighted – it is not apparent in pdf format but clearly visible after printing.

„DNAs for WG sequencing of six selected Y. enterocolitica strains were prepared using the

Genomic DNA Kit from Qiagen (QIAGEN ,Genomic DNA Buffer Set and Genomic-tip

100/G (QIAGEN Qiagen, Valencia, CA). Extracted DNAs were quantified using Qubit DNA

HS Quantitation kit (Invitrogen, Thermo Fisher Scientific, Waltham, MA) on a Qubit 3.0

fluorometer.” Please check if the state (CA, MA) shouldn’t be changed to USA

  • The sentence was corrected

Line 173: are the accession numbers available online? I had difficulties finding them

  • The genomes will be released on July 31, 2021 (see the notification letter from GenBank below) .

Line 199: I think it should be Y. enterocolitica (not the full name)

  • Corrected

Table 2: the numbers don’t add up in column „1997” (Apodemus spp.) – If I’m not mistaken

there should be „1” instead of „5”

  • Corrected

Figure 2: I think that the map of the year 1994 should be changed to 1993, because there were

no positive samples this year and in 1993 there was 1. Also, it would be better if table 2 and

figure 2 would include the same years

  • Corrected (the map is replaced).

Line 225: I advise to change „[…] of 53 of 572 […]” to „[…] in 53 out of 572 […]”

  • Changed

Table 3: name of bacteria should be in italics

  • Corrected

Line 284: this is the first occurrence of „E. coli” in the text so the full name should be used

  • The full name is provided

Line 294: „We were not be able to […]”

  • The typo is fixed

Line 356: „commensal l rats living closely to pig farms”

  • The typo is fixed

Page 12: correct citations (numbers instead of authors) and font size. Wrong format of citation

also in lines 159 and 428

  • Citations were corrected

Line 387: at the end of the line it should be respectively instead of „respectfully”

  • The word is replaced

Line 458: please check if the information included in lines 105-106 about the ethics

committee shouldn’t be included in this section

  • The Institutional Review Board (IRB) is established to protect the rights and welfare of human research subjects. The reported study does not include any investigation of people.

Line 469: please change „--„ to „—„

  • Changed

Line 505: no pages

  • The reference was replaced

Reference numbers 26, 30, 31: unnecessary hyperlink

  • Corrected

The email from GenBank is copied below:

From: genomes@ncbi.nlm.nih.gov <genomes@ncbi.nlm.nih.gov>

Sent: Thursday, April 8, 2021 8:43 PM

Subject: PGAP available Yersinia enterocolitica 18E17 065 D I Rint-95, Yersinia enterocolitica 18E17 065 D I Rint-91, Yersinia enterocolitica B18 Rint -231 (JAGKRV000000000-JAGKRX000000000)

Dear Ekaterine Zhgenti,

We are posting the PGAP output files to the submission portal (https://submit.ncbi.nlm.nih.gov/subs/genome) so that you may review the annotation.  There are 2 files under the "Details" or "detailed report" link: *bgpipe.output.sqn and *bgpipe.output.gb. Note that the PGAP annotation files are posted as they are finished, so files for some of the genomes listed above may not be present yet. They all should appear within a day.

The *bgpipe.output.sqn is the ASN.1 version of the file which can be viewed and/or edited in Genome Workbench. See: https://www.ncbi.nlm.nih.gov/tools/gbench and the tutorial

https://www.ncbi.nlm.nih.gov/tools/gbench/tutorial12/ or this webinar about viewing and editing a .sqn file, "A new way to prepare genome submissions using NCBI's Genome Workbench!" in

the archives https://www.ncbi.nlm.nih.gov/home/coursesandwebinars/ .

The *bgpipe.output.gb file is the text flatfile view. This format is for display only and cannot be edited.

ANY CHANGES MADE TO THE *bgpipe.output.gb FILE WILL NOT BE INCORPORATED INTO THE FINAL RECORD.

If you need to make extensive changes, you must edit the *bgpipe.output.sqn file.

The PGAP output files pass all automated validation tests required for data release. This does not mean that these files will be released without alteration. The PGAP output files have not yet been reviewed by our staff. Changes may be made to these output files during the final manual review step to fix sporadic problems that are difficult

to repair in a fully automated fashion. Additionally, if this submission is replacing a previously released version, the identifiers in the PGAP output files may change to ensure that the identifiers in the new version are distinct from the identifiers in the previously released version.

The release date is:

                Jul 31, 2021

Your genome(s) will be released on that date or when the GenBank Accession Number(s) or sequence data is publicly available, whichever is first.

NOTE: there is no need to renumber the locus_tag's; they are fine as they were produced by PGAP. If you wish to modify the author list, then just include the corrected list in your email; there is no need for you to modify the file for that change.

Please reply using the current Subject line.

Sincerely,

*******************************************************************

The GenBank Submissions Staff

Bethesda, Maryland USA

genomes@ncbi.nlm.nih.gov (replies/submission for WGS or complete genomes)

*******************************************************************